# Recent Vaccination Against SARS-CoV-2 Is Associated with Less Severe Disease in Working-Age Adults

**DOI:** 10.3390/ijerph21111501

**Published:** 2024-11-12

**Authors:** Kailey Fischer, Joanne M. Langley, Robyn Harrison, Samira Mubareka, Jeya Nadarajah, Marek Smieja, Louis Valiquette, Curtis Cooper, Jeff Powis, Allison McGeer, Brenda L. Coleman

**Affiliations:** 1Sinai Health, 600 University Ave, Toronto, ON M5G 1X5, Canada; kaf112@georgetown.edu (K.F.);; 2Canadian Center for Vaccinology, Dalhousie University, IWK and Nova Scotia Health, 5850 University Ave, Halifax, NS B3K 6R8, Canada; 3Division of Infectious Diseases, University of Alberta, 8440 112 St, Edmonton, AB T5J 3E4, Canada; 4Sunnybrook Health Sciences Centre, 2075 Bayview Ave, Toronto, ON M4N 3M5, Canada; 5Oak Valley Health, 381 Church St, Markham, ON L3P 13P, Canada; 6St. Joseph’s Healthcare, 50 Charlton Ave East, Hamilton, ON L8N 4A6, Canada; 7Centre Hospitalier Universitaire de Sherbrooke, 2500 Bd de l’université, Sherbrooke, QC J1K 2R1, Canada; 8Faculty of Medicine, University of Ottawa, 75 Laurier Ave E, Ottawa, ON K1N 6N5, Canada; 9Michael Garron Hospital, 825 Coxwell Avenue Toronto, ON M4C 3E7, Canada; 10School of Public Health, University of Toronto, Toronto, ON M5T 3M7, Canada; 11Laboratory Medicine and Pathobiology, University of Toronto, 1 King’s College Cir, Toronto, ON M5S 1A8, Canada

**Keywords:** COVID-19 vaccines, symptoms, illness, adult, Canada

## Abstract

Background: Essential workers, including those working in healthcare and education, are at higher risk of exposure to communicable diseases, including SARS-CoV-2. Reducing the rates of infection is important for their personal health and for the ongoing safe operation of essential services. Methods: Data from participants in two prospective cohort studies who tested positive for SARS-CoV-2 in 2020 through 2023 were used to determine whether vaccination against SARS-CoV-2 is associated with the severity of symptoms in working-age adults. Results: SARS-CoV-2-positive tests (N = 3757) were reported by 3093 participants (mean: 1.2 per person); 1229 (33%) illnesses did not interfere with regular activities, 1926 (51%) made participants too unwell for regular activities, and 602 (16%) required participant bed rest. Compared with vaccine receipt more than 12 months earlier, receipt within six months of an infection was associated with lower risk ratios for more severe illness (too unwell: 0.69 and bed rest: 0.67) compared with being able to conduct regular activities. More recent vaccination was also associated with lower odds reporting of systemic symptoms (fever, myalgia, arthralgia) and fewer solicited symptoms. Conclusion: Staying current with COVID-19 vaccinations should continue to be recommended since receiving a recent immunization lessened the severity of illness. Also, as symptoms of COVID-19 are now largely similar to other respiratory viruses, practitioners need to use this evidence to inform diagnostic testing and return-to-work policies.

## 1. Introduction

Public health measures to reduce the spread of severe acute respiratory syndrome coronavirus 2 (SARS-CoV-2) included requiring people to wear masks, to maintain a distance from others, and closing specific establishments (e.g., schools, restaurants). Nonetheless, many essential workers, including healthcare providers (HCPs) and education workers, were required to attend their workplace [1,2]. Due to the close contact that HCPs have with patients and education workers have with students, both populations are at risk of becoming ill with respiratory diseases, of transmitting the disease to others, and of leaving schools and healthcare institutions without qualified personnel. For this reason, once vaccines became available, staying up-to-date with COVID-19 vaccines was recommended or required in many healthcare and educational settings as well as being recommended for people at higher risk for severe illness [3,4].

Many researchers have reported waning immunity driven by the time between vaccination and/or infection and exposure and also by variants of SARS-CoV-2 that escape vaccine-induced immunity [5]. Wu et al. [6] conducted a meta-analysis of vaccine effectiveness (VE) against documented COVID-19 (any variant) in adults in which VE estimates following their primary series of vaccines dropped from 62% (95% confidence interval [CI] 53, 69) within 112–139 days of vaccination to 47% (CI 18, 65) at 280–307 days afterwards. Likewise, VE dropped from 56% (CI 35, 70) at 84–111 days to 43% (14, 62) at 112–139 days post-vaccination in this early review. In a second meta-analysis of adults, Song et al. [7] reported that the VE estimates against Omicron infections (symptomatic or not) following completion of a primary series dropped from 37.5% (CI 31.4, 43.1) within 180 days to 16.6% (CI 10.5, 22.3) for vaccinations received ≥90 to ≥200 days before test dates. Likewise, VE of a first booster dose dropped from 59.9% (CI 55.1, 64.1) to 31.5% (CI 22.7, 39.4) and VE for a second booster dropped from 59.6% (CI 52.0, 66.1) to 32.7% (CI 15.4, 46.4). Reinfections indicate that natural immunity is not life-long either, with protection against reinfection similarly waning over time [5,8,9].

Illnesses caused by SARS-CoV-2 vary in severity from asymptomatic infections to death [10]. Vaccination against SARS-CoV-2 prevents illness, helps reduce infection, but may also reduce illness severity if infected. In one American study, over 50% of symptomatic adult outpatients who tested positive for SARS-CoV-2 in 2022 reported problems completing their regular activities as well as being in pain (uncomfortable) and being anxious or depressed [11]. In that study, vaccinated participants reported faster recovery and less impact on their quality of life than unvaccinated individuals. A meta-analysis conducted in 2022 estimated that unvaccinated people were twice as likely to be infected, three times more likely to be hospitalized, and seven times more likely to be admitted to intensive care than vaccinated people in North America [12] indicating reduced severity with vaccination. Whether the severity of symptoms is reduced in non-hospitalized people is not well described. We hypothesize that recent vaccination will reduce the severity of symptoms in healthy working-age adults.

In this study, we aimed to determine whether recency of vaccination against SARS-CoV-2 was associated with the severity of symptoms of COVID-19 in working-age adults after adjusting for prior infection and demographic- and pandemic-specific variables. We also examined the association between (a) recency of vaccination and reporting of specific symptoms; (b) recency of vaccination and the total number of symptoms reported; (c) specific symptoms and severity of illness; and (d) specific symptoms by SARS-CoV-2 variant.

## 2. Methods

These analyses draw from two prospective cohort studies conducted in Canada. The COVID-19 Cohort Study followed HCPs from 19 participating acute care, rehabilitation, and complex care hospitals in Ontario (n = 13), Alberta (n = 3), Quebec (n = 1), and Nova Scotia (n = 2) as well as those who worked at private clinics in the Toronto area. HCPs were eligible if they were 18–75 years of age and were employed for ≥20 h per week [13]. The second study, the COVID-19 Cohort Study for teachers and education workers [14], enrolled people aged 18–75 years who were employed in any capacity by an Ontario school board for ≥8 h per week. Recruitment started in June 2020 for the HCPs and February 2021 for education workers and ended in June 2023 for both studies. Data collection ended upon withdrawal from either study or in December 2023, whichever occurred first. All participants provided written consent prior to enrolment.

Participants were eligible for this sub-study if they completed ≥80% of their first baseline questionnaire and reported at least one SARS-CoV-2-positive test during their participation. Eligibility was not restricted by vaccination status or previous infection with COVID-19.

### 2.1. Data Measurement

All questionnaires used for this sub-study were drafted and face validity assessed by the study investigators (epidemiology, infectious diseases, infection prevention and control) and then pilot-tested prior to use. Participants completed the questionnaires anonymously using secure online platforms.

Recency of vaccination, the independent variable for several of the objectives, was defined as having received the most recent dose of vaccine within six months of their illness. These data were derived from the vaccination questionnaire that collected names and dates of receipt of COVID-19 vaccines. Vaccination status ten days prior to the date of the illness was used for these analyses to allow time for antibodies to develop.

Participants were asked to complete illness reports if they had a COVID-19 test and/or symptoms compatible with a COVID-19 illness. Infection was defined as an illness in association with either a polymerase chain reaction (PCR) or rapid antigen test that was positive for SARS-CoV-2. Illnesses were excluded from these analyses if the participant reported an eligible positive test (a) within the previous 30 days or (b) that was followed by a negative result within 48 h.

Illness questionnaires provided information on the severity of each illness, the dependent variable for two objectives. The possible responses to the question included “I was able to do regular activities throughout the illness”, “I was not well enough to do regular activities”, “I was not well enough to get out of bed”, and “I was admitted to hospital”. Due to small numbers, those admitted to hospital were combined with those too unwell to get out of bed.

Solicited symptoms that were used for various independent and dependent variables in these analyses were also derived from the illness reports. Solicited data on symptoms experienced (present/absent) included cough (new/worsening), dyspnea (new), chest pain/pressure/heaviness, fever, feeling generally unwell, fatigue (new onset), confusion (new onset), myalgia (new onset), arthralgia (new onset), ear ache/infection, headache (unusual or long lasting), coryza, sinus pain, sore/scratchy throat or laryngitis, anorexia (new onset), nausea and/or vomiting, diarrhea, ageusia (new onset), and anosmia (new onset). The total number of solicited symptoms, the independent variable in one objective, is simply the sum of the symptoms reported by the participant.

All participants were asked to complete a baseline questionnaire at enrolment and again every 12 months (for HCPs) or again every September (for education workers); they captured personal, household, and workplace-related data. Baseline data from the questionnaire most recently preceding the date of illness were used to measure potentially confounding variables. Health status was assessed by asking “In general, would you say your health is…excellent, very good, good, fair/poor” [15]. Participants were also asked “Have you ever been diagnosed by a health profession with [each of]… asthma, chronic obstructive pulmonary or other chronic lung disease, diabetes, heart disease, cancer treated in the past 5 years, liver/kidney disease, HIV/AIDS or other immune suppressing disease, or other long-term or chronic condition that has lasted, or is expected to last, at least 6 months?” Smoking status questions included “In your lifetime, have you smoked a total of 100 or more cigarettes (about 4 packs), 50 cigars, or 50 pipes full of tobacco?”, and if they replied yes, “Do you currently smoke tobacco… daily, occasionally, or not at all”. Daily and occasionally were combined to describe current smoking status.

The SARS-CoV-2 variant causing most COVID-19 in Canada at the time of the participant’s infection was used to assign the predominant variant for analysis [16] and used as the dependent variable for one objective.

### 2.2. Data Analysis

Categorical factors were compared using Chi square or Fischer’s exact tests. Continuous variables were compared using t-tests or Wilcoxon rank sum or K-sample equality-of-medians tests, as appropriate. All tests of statistical significance were two-sided with significance set at *p* < 0.05.

Multivariable logistic regression, clustered by individual identifier (to account for repeated infections), adjusted for study (HCP vs. educator), and employing robust variance estimation was used to estimate the association between vaccination and self-reported levels of severity of illness: able to perform regular activities (reference group), too unwell to perform regular activities, or required bed rest (six cases who were admitted to hospital were re-categorized as requiring bed rest). Previous COVID-19 infection and other potentially confounding variables (age, gender, predominant circulating strain of SARS-CoV-2, health status, chronic underlying conditions, and smoking status) were removed sequentially from the model starting with those with the highest *p*-value until all remaining variables were associated at a *p*-value of ≤0.10. Factors removed from the saturated model were added back into the reduced model, one at a time, to assess their level of association and their impact on the other estimates. If the removal or addition of any variable changed the estimates of other variables by >10%, they were retained. Models were assessed for potential effect measure modification for biologically plausible pairings (e.g., age and health status). For highly correlated variables, the variable associated with the highest Wald statistic in the final models was reported. We were unable to include the four observations for which the participant identified as “other” gender due to the associated instability of estimates. Sensitivity analyses were conducted to determine whether the associations between vaccination timing and illness severity were affected by (a) previous infection(s) and (b) asymptomatic infections; the same methods described above were used. A post hoc analysis stratified by cohort was also conducted.

To assess the association between recency of vaccination (<6 months versus >12 months) and specific symptoms reported, logistic regression, adjusted for confounding and repeat infections by the same participant, were conducted.

The association between recency of vaccination and the total number of solicited symptoms reported during the illness was examined using linear regression models and the same methods described above.

The power to detect a significant difference between the smaller (bedrest) and referent groups is 100% using a type 1 error probability of 0.05. Similarly, the power to detect significant differences in the association of vaccination timing and (a) each symptom and (b) total number of symptoms was 100% using a type 1 error of 0.05. Analyses were conducted in Stata SE v18 [17].

## 3. Results

Among the 1382 healthcare and 1711 education workers participating in the studies, 2717/3093 (87.8%) were female, the mean age was 44.6 years (IQR 37, 52), and 2268 (73.3%) reported being in very good or excellent health. One positive SARS-CoV-2 test was reported by 3093 participants (82.3%), two by 607, three by 53, and four by 4.

As shown in Table 1, 1229 of the 3757 infections (32.7%) were mild enough that the participant was able to do regular activities throughout their illness (123 reported no symptoms), 1926 (51.3%) were not able to do regular activities (unwell), and 602 (16.0%) were not well enough to get out of bed for part of their illness (including six hospitalizations). The majority of reported cases occurred during the Omicron BA.4/BA.5 period.

Results of multivariable logistic regression (see Table 2) indicate that, compared with being vaccinated >12 months before the infection, those infected within 6 months of vaccination were significantly less likely to report being too unwell to do regular activities (relative risk ratio [RRR] 0.69; 95% CI 0.51, 0.92) or require bed rest (RRR 0.67; 95% CI 0.46, 0.98). The risk of being unable vs. able to conduct regular activities was not significantly different when vaccinated 6–12 months vs. >12 months prior to the infection. A post hoc analysis, stratified by study cohort, was conducted; it showed no substantial differences in the results, which is not unexpected given the similarity in ages and gender.

As shown in Appendix A, having had a previous infection, although statistically significant, did not have a substantive impact on the estimated effect of vaccination within 6 months compared with >12 months before infection: RRR 0.72 (95% CI 0.54, 0.97) for being too unwell compared with being able to do regular activities; however, the estimate for needing bed rest was no longer significant (RRR 0.71; 95% CI 0.49, 1.03).

A second sensitivity analysis that excluded the 123 asymptomatic infections (see Appendix A) resulted in similar estimates of the association as the full data set; the RRR was 0.69 [95% CI 0.51, 0.93] for too unwell and 0.67 [95% CI 0.45, 0.98] for needing bed rest compared to being able to conduct regular activities for people who were vaccinated within 6 months compared with >12 months before infection. Similar to the full data set results, the risk of being unable to do regular activities was similar whether vaccinated 6–12 months or >12 months prior to infection.

### Symptoms

Infections for which the participants were vaccinated within the previous 6 months were significantly less likely to have reports of myalgia, headache, arthralgia, or fever than infections in which the vaccination was more than one year prior to onset. As shown in Table 3, there were no significant differences noted for other symptoms.

Compared with infections in which participants were vaccinated >12 months earlier (see Appendix A), those in which the receipt of a COVID-19 vaccine was within six months of infection had about 60% fewer symptoms reported (−0.58; 95% CI −1.02, −0.14).

Appendix A provides the frequency of reported symptoms by severity of illness. The most frequently reported symptoms included coryza (71.4%), sore throat/laryngitis (68.8%), feeling generally unwell (64.7%), and cough (63.6%). Ageusia and/or anosmia was reported for 17.4% of illnesses, while fever was reported for 25.0%.

As shown in Table 4, the percent of illnesses with specific symptoms reported varied by the predominant variant. The median number of symptoms reported per illness was 6 (IQR 3, 9; range 0 to 14), with fewer reported during the first waves of the pandemic (Wuhan through Omicron BA.1) than during the later waves (Omicron BA.2 through Recombinant to December 2023).

## 4. Discussion

This longitudinal study of Canadian adults working in the essential services of healthcare or education has data for COVID-19 infections from 2020 to 2023, covering a range of SARS-CoV-2 variants. It is one of few studies reporting symptoms from adults who were infected, but not hospitalized, due to COVID-19. It found that infections for which the participant reported receipt of a COVID-19 vaccine within the previous six months were significantly less likely to have symptoms that made them too unwell to continue with their regular activities or to require bed rest when compared with infections for which vaccinations were received more than twelve months earlier. Receipt of a vaccine within six months of the reported infection was also associated with fewer solicited symptoms than if vaccination was more than one year before the infection.

These findings are similar to those reported by others. Adults living in the United Kingdom who received two doses of COVID-19 vaccine had significantly lower odds of having symptoms that lasted ≥28 days, were less likely to report ≥5 symptoms in the first week, and were less likely to visit the hospital between December 2020 and July 2021 [18]. Boulware and colleagues reported that vaccinated overweight or obese adults 30–85 years of age reported less severe symptoms that resolved more quickly than unvaccinated participants [19]. In that study of overweight adults, participants who had received at least one booster dose reported the least severe symptoms and, unlike those who received only the primary series, maintained the benefit over unvaccinated people through to January 2022. Also, Di Fusco and colleagues reported that American adult outpatients who had breakthrough infections after receiving a bivalent BA.4/5 vaccine reported fewer acute symptoms and, after the first week, reported fewer hours of lost work than unvaccinated people [11].

In line with studies reporting waning immunity over time [6,7], the results of this study indicate that infections in which participants had been vaccinated within the previous six months were significantly less likely to include the systemic symptoms of fever, arthralgia, myalgia, or headache compared with those for which the most recent vaccine was >12 months prior. Bramante et al. [20] reported that vaccinated participants with illnesses in 2021 were significantly less likely to report fever/feverishness, arthralgia, myalgia, diarrhea, or nausea than unvaccinated participants. During the Omicron-predominant period in Japan, vaccinated, symptomatic adults with COVID-19 were significantly less likely to report fever, arthralgia or myalgia, diarrhea, severe fatigue, headache, anosmia and/or ageusia, or dyspnea within five days of symptom onset than unvaccinated adults [21]. Reduction in systemic symptoms despite waning immunity as measured by neutralizing antibodies supports the findings that protection from severe disease is likely mediated by cellular immunity [22].

In this study, symptomatic infections reported during periods predominated by different SARS-CoV-2 variant periods had somewhat different symptoms reported by participants. Those occurring during the periods dominated by infections caused by the Wuhan, Alpha, and Delta variants were significantly more likely to have reports of sore throat/laryngitis, feeling generally unwell, cough, fatigue, sinus pain, and anosmia and/or ageusia compared with the Omicron and recombinant periods. Anosmia and cough were also more commonly reported among vaccinated participants during the Omicron compared with the Delta period in non-hospitalized adults in the United Kingdom [23]. Like the results of this study, others have also reported higher incidence of coryza when infected with later variants [11,20,24]. Nakakubo and colleagues found that most participants reported cough, sore throat, and nasal discharge, symptoms common to many respiratory viruses, during the Omicron BA.2 and BA.5 predominant periods in Japan [21]. Geismar et al. also found that the symptomatology of SARS-CoV-2 in the United Kingdom had shifted to be more similar to other respiratory viruses [25]. The changes in the symptomatology associated with different SARS-CoV-2 variants over time highlight the need for clinicians, public health officials, and the general public to consider the possibility of COVID-19 in the absence of symptoms that some may have previously assumed differentiated the illness from other respiratory diseases. It underscores the importance of symptom review to inform diagnostic testing for infection prevention and control and return to work after illness policies, where applicable.

There are limitations to this sub-study, as participants self-reported all data introducing the possibility of missing data, especially for milder illnesses. Both populations in these studies, healthcare and education workers, were highly vaccinated once vaccines were available, minimizing the number of unvaccinated participants. With regard to assessment of severity of illness, since our studies included asymptomatic cases it may represent a more complete picture of the spectrum of COVID-19 infection than reports limited to hospitalized and/or symptomatic people. We did not follow participants to assess longer term outcomes such as long COVID.

## 5. Conclusions

Recent vaccination against COVID-19 reduces the reported severity of illness in working-age adults who test positive for SARS-CoV-2. Vaccination within the previous six months was associated with a lower likelihood of systemic symptoms including fever, myalgia, and arthralgia; symptoms that were associated with being too unwell to conduct regular activities. Staying current with vaccinations should continue to be recommended, especially for people at high risk of exposure and transmission to others, and where illness could have an impact on essential operations such as education and healthcare. Also, as symptoms of COVID-19 are now largely similar to other respiratory viruses, practitioners need to use this evidence to inform diagnostic testing and return to work policies.

## Figures and Tables

**Table 1 ijerph-21-01501-t001:** Characteristics of respondents at the time of each reported positive SARS-CoV-2 test; Canadian healthcare and education workers, June 2020–December 2023. Number (percent) unless otherwise noted.

Characteristics	Regular Activities (N = 1229)	Too Unwell for Activities (N = 1926)	Bedrest (N = 602)	Total Cases (N = 3757)
COVID-19 vaccine before infection				
None/one ^1^	72 (5.9)	81 (4.2)	37 (6.2)	190 (5.1)
>12 months prior	129 (10.5)	281 (14.6)	93 (15.5)	503 (13.4)
6 to 12 months prior	367 (29.9)	624 (32.4)	193 (32.1)	1184 (31.5)
<6 months prior	661 (53.8)	940 (48.8)	279 (46.4)	1880 (50.0)
Previous COVID-19 infection				
None	972 (79.1)	1544 (80.2)	486 (80.7)	3002 (79.9)
>12 months prior	134 (10.9)	231 (12.0)	76 (12.6)	441 (11.7)
6 to 12 months prior	80 (6.5)	119 (6.2)	27 (4.5)	226 (6.0)
<6 months prior	43 (3.5)	32 (1.7)	13 (2.2)	88 (2.3)
Gender: Female	1035 (84.2)	1716 (89.1)	558 (92.7)	3309 (88.1)
Male	194 (15.8)	210 (10.9)	44 (7.3)	448 (11.9)
Age, in years, median (IQR)	43 (36, 51)	45 (38, 52)	47 (39, 53)	45 (37, 52)
SARS-CoV-2 variant period				
Wuhan (L)	42 (3.4)	38 (2.0)	19 (3.2)	99 (2.6)
Alpha	21 (1.7)	28 (1.5)	12 (2.0)	61 (1.6)
Delta	17 (1.4)	8 (0.4)	1 (0.2)	26 (1.0)
Omicron BA.1	319 (26.0)	383 (20.0)	81 (13.5)	783 (20.8)
Omicron BA.2	238 (19.4)	420 (21.8)	143 (23.8)	801 (21.3)
Omicron BA.4/5	385 (31.3)	625 (32.5)	201 (33.4)	1211 (32.2)
Recombinant	207 (16.8)	424 (22.0)	145 (24.1)	776 (20.6)
Study: Healthcare	612 (49.8)	862 (44.8)	177 (29.4)	1651 (43.9)
Education	617 (50.2)	1064 (55.2)	425 (70.6)	2106 (56.1)
Health status				
Poor/fair/good	266 (21.6)	538 (27.9)	202 (33.6)	1006 (26.8)
Very good	619 (50.4)	908 (47.1)	285 (47.3)	1812 (48.2)
Excellent	344 (28.0)	480 (24.9)	115 (19.1)	939 (25.0)
Asthma and/or COPD	189 (15.4)	337 (17.5)	122 (20.3)	629 (16.7)
Diabetes	42 (3.4)	45 (2.3)	26 (4.3)	113 (3.0)
Heart disease	15 (1.2)	32 (1.7)	18 (3.0)	65 (1.7)
Cancer in the past 5 years	15 (1.2)	29 (1.5)	13 (2.2)	57 (1.5)
Liver or kidney disease	11 (0.9)	13 (0.7)	8 (1.3)	32 (0.8)
Immunosuppressed ^2^	44 (3.6)	65 (3.4)	24 (4.0)	133 (3.5)
Smoking status				
Never	932 (75.8)	1482 (76.9)	450 (74.8)	2864 (76.2)
Former	248 (20.2)	358 (18.6)	135 (22.4)	741 (19.7)
Current	49 (4.0)	86 (4.5)	17 (2.8)	152 (4.1)

^1^ One dose of a two-dose vaccine (N = 22/190). ^2^ Immunosuppressive medication or HIV/AIDS. Note that the number of reported cases, rather than the number of participants, is presented to accurately report factors that changed for participants with >1 infection (e.g., vaccination status).

**Table 2 ijerph-21-01501-t002:** Association between recency of COVID-19 vaccination and severity of illness; Canadian healthcare and education workers, June 2020–December 2023. Relative risk ratio ^1^ (95% confidence interval).

Variable	Regular Activities (N = 1229)	Unwell (N = 1926)	Bed Rest (N = 602)
COVID-19 vaccine ^2^	Reference group		
>12 months prior		Referent	Referent
None/one of two dose		0.79 (0.40, 1.58)	0.91 (0.33, 2.54)
6 to 12 months prior		0.81 (0.62, 1.07)	0.80 (0.55, 1.14)
<6 months prior		0.69 (0.51, 0.92) *	0.67 (0.46, 0.98) *
Confounding variables			
Previous COVID-19 ^2^	Reference group		
>12 months prior		Referent	Referent
None		1.21 (0.93, 1.59)	1.27 (0.88, 1.84)
6 to 12 months prior		0.94 (0.65, 1.35)	0.65 (0.38, 1.10)
<6 months prior		0.50 (0.30, 0.84) *	0.65 (0.33, 1.29)
Age in years	Reference group	1.01 (1.00, 1.02)	1.01 (1.00, 1.02)
Gender: Female	Reference group	Referent	Referent
Male		0.65 (0.52, 0.81) †	0.40 (0.28, 0.58) †
Health status: Poor/fair/good	Reference group	Referent	Referent
Very good		0.76 (0.63, 0.91) *	0.68 (0.53, 0.86) *
Excellent		0.72 (0.58, 0.89) *	0.49 (0.36, 0.66) †
Heart disease: No	Reference group	Referent	Referent
Yes		1.24 (0.68, 2.25)	2.17 (1.06, 4.47) *
SARS-CoV-2 variant period	Reference group		
Wuhan (L)		Referent	Referent
Alpha		1.56 (0.75, 3.23)	1.22 (0.50, 2.99)
Delta		0.66 (0.21, 2.09)	0.21 (0.02, 2.16)
Omicron BA.1		1.47 (0.66, 3.24)	0.68 (0.22, 2.05)
Omicron BA.2		2.24 (1.01, 4.99) *	1.68 (0.55, 5.17)
Omicron BA.4/5		1.88 (0.83, 4.22)	1.29 (0.42, 4.00)
Recombinant		2.19 (0.95, 5.02)	1.54 (0.48, 4.93)

* *p* ≤ 0.05; † *p* ≤ 0.001. ^1^ Adjusted for study (healthcare vs. education worker), >1 illness from the same participant, and other variables in column. ^2^ ≥10 days prior to date of [current] infection.

**Table 3 ijerph-21-01501-t003:** Odds ratios for individual symptoms for infections in which participants were vaccinated <6 vs. >12 months earlier; Canadian healthcare and education workers, June 2020–December 2023. Odds ratio ^1,2^ (95% confidence interval).

Symptom	Odds Ratio ^1,2^	*p*-Value
Myalgia (new onset)	0.66 (0.51, 0.86)	0.002
Headache (unusual or long lasting)	0.68 (0.53, 0.87)	0.003
Arthralgia (new onset)	0.69 (0.50, 0.95)	0.02
Fever	0.72 (0.54, 0.95)	0.02
Confusion (new onset)	0.55 (0.29, 1.01)	0.06
Diarrhea	0.74 (0.52, 1.06)	0.10
Ageusia and/or anosmia (new onset)	0.78 (0.56, 1.08)	0.13
Generally unwell	0.81 (0.61, 1.06)	0.13
Anorexia (new onset)	0.83 (0.62, 1.11)	0.20
Fatigue (abnormal)	0.86 (0.66, 1.10)	0.23
Sore/scratchy throat/laryngitis	1.17 (0.89, 1.55)	0.25
Cough (new onset)	0.86 (0.66, 1.12)	0.27
Ear ache/ear infection	0.82 (0.56, 1.18)	0.28
Nausea/vomiting	0.83 (0.54, 1.27)	0.40
Sinus pain/pressure	0.89 (0.68, 1.17)	0.42
Chest pain/pressure/heaviness	1.14 (0.78, 1.64)	0.50
Coryza	1.08 (0.81, 1.45)	0.58
Dyspnea (new onset)	0.92 (0.66, 1.28)	0.61

^1^ Vaccinated <6 months compared with >12 months before infection. ^2^ Each estimate was adjusted for previous infection, age, gender, health status, heart disease, variant period, study, and repeat infection by same participant.

**Table 4 ijerph-21-01501-t004:** Symptoms reported by predominant SARS-CoV-2 variant period; Canadian healthcare and education workers, June 2020–December 2023. Number (percent) unless otherwise stated.

Symptom	Wuhan (L) (N = 99)	Alpha (N = 61)	Delta (N = 26)	Omicron BA.1 (N = 783)	Omicron BA.2 (N = 801)	Omicron BA.4/5 (N = 1211)	Recomb-inant ^1^ (N = 776)	*p*-Value
Coryza	38 (38.4)	28 (45.9)	19 (73.1)	538 (68.7)	581 (72.5)	873 (72.1)	604 (77.8)	<0.001
Sore throat/laryngitis	32 (32.3)	25 (41.0)	10 (38.5)	515 (65.8)	614 (76.7)	843 (69.6)	547 (70.5)	<0.001
Generally unwell	54 (54.6)	34 (55.7)	13 (50.0)	454 (58.0)	541 (67.5)	789 (65.2)	546 (70.4)	<0.001
Cough	33 (33.3)	27 (44.3)	12 (46.2)	488 (62.3)	543 (67.8)	811 (67.0)	475 (61.2)	<0.001
Fatigue	49 (49.5)	22 (36.1)	9 (34.6)	364 (46.5)	456 (56.9)	669 (55.2)	461 (59.4)	<0.001
Headache	37 (37.4)	35 (57.4)	12 (46.2)	409 (52.2)	427 (53.3)	650 (53.7)	447 (57.6)	0.01
Myalgia	46 (46.5)	24 (39.3)	9 (34.6)	325 (41.5)	345 (43.1)	548 (45.3)	398 (51.3)	0.003
Sinus pain	18 (18.2)	11 (18.0)	5 (19.2)	208 (26.6)	253 (31.6)	365 (30.1)	265 (34.2)	0.001
Anorexia	26 (26.3)	13 (21.3)	5 (19.2)	141 (18.0)	186 (23.2)	284 (23.5)	210 (27.1)	0.004
Fever	30 (30.3)	16 (26.2)	3 (11.5)	156 (19.9)	196 (24.5)	333 (27.5)	205 (26.4)	0.003
Arthralgia	22 (22.2)	10 (16.4)	1 (3.9)	139 (17.8)	135 (16.9)	234 (19.3)	161 (20.8)	0.15
Dyspnea	21 (21.2)	12 (19.7)	3 (11.5)	120 (15.3)	137 (17.1)	200 (16.5)	140 (18.0)	0.62
Anosmia and/or ageusia	40 (40.4)	22 (36.1)	9 (34.6)	109 (13.9)	102 (12.7)	233 (19.2)	137 (17.6)	<0.001
Diarrhea	16 (16.2)	8 (13.1)	2 (7.7)	81 (10.3)	94 (11.7)	145 (12.0)	123 (15.9)	0.03
Ear ache	5 (6.1)	2 (3.9)	2 (7.7)	81 (10.3)	102 (12.7)	149 (12.3)	106 (13.7)	0.09
Chest pain	15 (15.2)	13 (21.3)	0 (0.0)	100 (12.8)	134 (16.7)	173 (14.3)	96 (12.4)	0.02
Nausea	10 (10.1)	8 (13.1)	2 (7.7)	76 (9.7)	97 (12.1)	146 (12.1)	85 (11.0)	0.68
Confusion	6 (6.1)	3 (4.9)	0 (0.0)	24 (3.1)	39 (4.9)	55 (4.5)	42 (5.4)	0.29
Number reported, median (IQR)	5 (1.9)	5 (2.8)	4 (2.8)	5 (3.8)	6 (4.8)	6 (3.9)	6 (4.9)	0.001

IQR: interquartile range. ^1^ Follow up ends in December 2023.

## Data Availability

The data that support the findings of this study are available on request from the corresponding author, [BLC]. The data are not publicly available due to information that could compromise the privacy of research participants.

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
