# Peer review of "Recent Vaccination Against SARS-CoV-2 Is Associated with Less Severe Disease in Working-Age Adults"

_ijerph, 2024, doi:10.3390/ijerph21111501_

Round 1
Reviewer 1 Report (Previous Reviewer 1)
Comments and Suggestions for Authors
This version has already addressed my previously raised concerns. However, I noticed some typographical errors in the supplementary materials, specifically in Table 5S, that need to be revised for consistency throughout the manuscript.
"Alternative Terminology To maintain consistency with medical terminology, I recommend using specific clinical terms instead of lay language. This would align better with other variable names mentioned in the manuscript: Anosmia instead of "loss of sense of smell" Ageusia instead of "loss of sense of taste" Anorexia instead of "loss of appetite" Rhinorrhea and nasal congestion instead of "runny nose" and "stuffy nose" Dyspnea instead of "shortness of breath"
Response: We have made these changes."
Author Response
Thank you for your time and expertise. We appreciate it very much.
This version has already addressed my previously raised concerns. However, I noticed some typographical errors in the supplementary materials, specifically in Table 5S, that need to be revised for consistency throughout the manuscript.
Response: My goodness. Thank you. We totally missed that! A revised supplementary document is attached!
"Alternative Terminology To maintain consistency with medical terminology, I recommend using specific clinical terms instead of lay language. This would align better with other variable names mentioned in the manuscript: Anosmia instead of "loss of sense of smell" Ageusia instead of "loss of sense of taste" Anorexia instead of "loss of appetite" Rhinorrhea and nasal congestion instead of "runny nose" and "stuffy nose" Dyspnea instead of "shortness of breath"
Response: We have made these changes. Thank you very much"
Reviewer 2 Report (New Reviewer)
Comments and Suggestions for Authors
Dear Authors,
I commend you for conducting this straightforward research on the association between COVID-19 symptoms and severity and the time since the last vaccine dose. Beside the main findings, it’s noteworthy that vaccination beyond 12 months appears to have a similar effect as not being vaccinated. I have some comments as follows:
1. There are track changes in the manuscript which should be resolved prior to submission.
2. Line 74-75, please revise this sentence: “Although it follows would Whether the severity of symptoms is reduced in non-hospitalized people is not well described.”
3. Line 103-113: Align the symptom terminology with tables 2 and 3: loss of sense of taste/ agnosia, loss of sense of smell/ anosmia, loss of appetite/ anorexia, runny or study nose/ coryza, and shortness of breath/ dyspnea
4. In the results section, please describe the prevalence of different variants.
5. In table 1, I suggest adding an analysis stratified by being healthcare or education workers.
6. It’s unclear if the observed differences are due to the number of doses received or the time between vaccination and infection. If the latter, could this imply that COVID-19 vaccines should be administered every 6 months? This seems particularly relevant given the results in table 1, which suggest no difference between not being vaccinated and having been vaccinated more than 12 months ago.
Author Response
Thank you for your considered review. We appreciate your time and expertise in making this manuscript a better product.
- There are track changes in the manuscript which should be resolved prior to submission.
Thank you. We have done this.
- Line 74-75, please revise this sentence: “Although it follows would Whether the severity of symptoms is reduced in non-hospitalized people is not well described.”
Thank you. We dropped the first 4 words…
- Line 103-113: Align the symptom terminology with tables 2 and 3: loss of sense of taste/ agnosia, loss of sense of smell/ anosmia, loss of appetite/ anorexia, runny or study nose/ coryza, and shortness of breath/ dyspnea
We have updated the data measurement section to match the terms used in the tables, thanks.
- In the results section, please describe the prevalence of different variants.
Another reviewer asked us to move Table S1 to the main text. In doing so, the number of cases during each period are presented. We have also added a line to the text (see line 185).
- In table 1, I suggest adding an analysis stratified by being healthcare or education workers.
The reason we didn’t stratify by cohort was because both cohorts are largely female and of a similar age. We saw no reason to think that their symptoms should be different. However, we ran the analysis by cohort just to allay your concerns. There was no difference by cohort. We have added this to the methods (line 163-164) and results (lines 200-203).
- It’s unclear if the observed differences are due to the number of doses received or the time between vaccination and infection. If the latter, could this imply that COVID-19 vaccines should be administered every 6 months? This seems particularly relevant given the results in table 1, which suggest no difference between not being vaccinated and having been vaccinated more than 12 months ago.
We cannot present the data by the number of doses received because the number of doses available changed over the course of the follow-up (June 2020 through December 2023); the only way to present the data would be by variant period and this would severely reduce the power to find associations. This is the reason we chose to present the data by recency of vaccination rather than number of vaccines received. The conclusions are, as you say, that vaccination within the previous 6 months reduced the severity of symptoms in working age adults.
Thank you!
Reviewer 3 Report (New Reviewer)
Comments and Suggestions for Authors
The fieldwork section is very well planned, but the statistical analyses have been conducted incorrectly.
Introducion: literature review in the study is sufficient. However, the study's hypothesis has not been provided, and there is no contribution of the study will make to the existing literature.
Methods:
-
The response variable “Severity of illness was a four-category response variable” needs clarification. What sources were used in developing these questions? Has a validation study been conducted for these questions?
-
The sample size or the power of the study should be mentioned.
-
The definition of "very good or excellent health" is unclear.
-
There is no mention of the normal distribution properties.
-
The dependent and independent variables of the study have not been clearly defined.
-
It appears that there is confusion and incorrect interpretation of the concepts of linear vs. logistic regression, as well as odds ratios (OR) and relative risks (RR), particularly in Table 1. The study claims to have calculated relative risks, but logistic regression was performed, indicating confusion in both the concepts and analyses. A multivariable model refers to a regression model that includes multiple independent variables (predictors) to explain or predict an outcome. This is commonly used in various types of regression, including logistic regression (for binary outcomes) and linear regression (for continuous outcomes). However, the results in the study are presented using multinomial logistic regression analysis, while Table S4 refers to multiple linear regression.
-
Table S1 should be presented in the main text, not as supplementary material.
Author Response
Thank you for your thoughtful review. We think that your suggestions have help make the methods more clear for the readers.
Introducion: literature review in the study is sufficient. However, the study's hypothesis has not been provided, and there is no contribution of the study will make to the existing literature.
Thank you. We have added a hypothesis, as suggested.
Methods:
- The response variable “Severity of illness was a four-category response variable” needs clarification. What sources were used in developing these questions? Has a validation study been conducted for these questions?
Thank you. We have added information in lines 105-108 that the questions were drafted by, and had the face validity checked by, the investigators and pilot tested prior to use and that all items are self-reported.
Lines 108-112 are from the previous version; they describe the question and responses for the severity of illness.
Further, we have added details about other questions used in the analyses to lines 123-133.
- The sample size or the power of the study should be mentioned.
We have added a power calculation to the analysis section. See lines 170-173.
- The definition of "very good or excellent health" is unclear.
Health status was self-reported.
- There is no mention of the normal distribution properties.
We have added the phrase “as appropriate” to indicate that we did, indeed, check for normality prior to determining whether to report means or medians.
- The dependent and independent variables of the study have not been clearly defined.
It appears that there is confusion and incorrect interpretation of the concepts of linear vs. logistic regression, as well as odds ratios (OR) and relative risks (RR), particularly in Table 1. The study claims to have calculated relative risks, but logistic regression was performed, indicating confusion in both the concepts and analyses. A multivariable model refers to a regression model that includes multiple independent variables (predictors) to explain or predict an outcome. This is commonly used in various types of regression, including logistic regression (for binary outcomes) and linear regression (for continuous outcomes). However, the results in the study are presented using multinomial logistic regression analysis, while Table S4 refers to multiple linear regression.
Of the three different analyses in this paper:
The multivariable logistic regression analysis was to determine the association between vaccination and severity of illness (see lines 145-148). We used Stata to complete these analysis and it provides relative risk ratios. This indicates the risk of being in one group (bedrest, for example) compared to the referent group (able to do regular activities, in our study). As stated in the header of Table 2 and line 189 of the results, we are reporting relative risk ratios.
Next, we used regular logistic regression (thus, reported odds ratios) to assess the association between the recency of vaccination and symptom reporting (yes/no for each solicited symptom) as indicated in the methods (lines 164-166) and in Table 3.
Finally, we used linear regression to assess the association between the recency of vaccination and the number of symptoms reported (line 167-169 of the methods and lines 233-236 of the results and in Table S3).
We split the paragraphs by analysis to make the distinction easier for readers thanks to your review. Also, (line 164-165) we added the comparison of the vaccination periods to help clarify. In line 167, we added the word “total” to help clarify for the total number of symptoms variable.
- Table S1 should be presented in the main text, not as supplementary material.
We have put Table S1 in the main text as Table 1.
We thank you for your time.
Round 2
Reviewer 3 Report (New Reviewer)
Comments and Suggestions for Authors
There are still gaps in the methodology;
Lines 105-134: It is stated that a "Questionnaire" was used in the study and its validity was established. It should be mentioned which literature sources were used to create the questionnaire (related references should be given), how its validity was determined, and what the validity results are If done.
The dependent and independent variables of the study should be described with 1-2 sentences in the methodology before statistical analysis section.
Author Response
There are still gaps in the methodology;
Lines 105-134: It is stated that a "Questionnaire" was used in the study and its validity was established. It should be mentioned which literature sources were used to create the questionnaire (related references should be given), how its validity was determined, and what the validity results are If done.
Validation was not completed beyond what was reported in the manuscript. Questions for this manuscript were developed by the investigators although the Canadian Community Health Survey may have been source for one of them. As such, we cited that source.
The dependent and independent variables of the study should be described with 1-2 sentences in the methodology before statistical analysis section.
Thank you. I have tried to clarify the data analysis section (and the objective at the end of the intro) to make this easier for readers. Your suggestions have been invaluable for clearing this up.
This manuscript is a resubmission of an earlier submission. The following is a list of the peer review reports and author responses from that submission.
Round 1
Reviewer 1 Report
Comments and Suggestions for Authors This manuscript explores the impact of recent COVID-19 vaccination on reducing the likelihood of severe COVID-19 among working-age adults. It serves as an extension of outcome analysis from a previous publication. While the methodology is robust and well-executed, the manuscript's content, particularly in the Introduction and Discussion sections, is somewhat underdeveloped. To elevate this work to the level of an original article, I recommend enhancing these sections with additional information and stronger arguments, making it more comprehensive and informative rather than resembling a research letter or note. Major Concerns.1. Confounding Variables: Smoking Status
Did this study collect smoking status (never smoked, former smoker, current smoker)? Smokers have a higher risk of severe infection compared to those who have never smoked. This could be an important confounding variable to consider in the analysis. 2. Confounding Variables: BMI Was BMI data collected, or at least a dichotomous category such as non-obesity vs. obesity? Obese individuals have a significantly higher risk of severe infection compared to those with normal weight. Including this as a confounding factor would strengthen the analysis. 3. Immunity Waning and Booster Vaccination The manuscript focuses on how recent COVID-19 vaccinations reduce the severity of infection, but the Introduction and Discussion sections lack depth regarding causes of severe infection, such as "immunity waning." It would be beneficial to include more evidence and discussion on the role of waning immunity and booster vaccination. I recommend expanding these sections using examples from relevant publications: https://www.nejm.org/doi/full/10.1056/NEJMoa2115481
https://www.frontiersin.org/journals/public-health/articles/10.3389/fpubh.2023.1342118/full Minor Concerns
1. Vaccination Status:
Please clarify the vaccination criteria in the Methods section. Did all participants receive at least two doses or complete the primary vaccination series?
It’s important that readers are not required to refer back to your previous publication to understand the major inclusion criteria for this study. Comments
1. Subgroup Analysis by Age
Have you considered conducting a subgroup analysis based on age? This could involve categorizing participants by every 10 years or by life stages (e.g., young adults, middle-aged adults, elderly). Alternatively, you could use a simpler dichotomy, such as non-elderly vs. elderly, depending on your outcome analysis.
Since older adults tend to experience more severe COVID-19 infections compared to younger individuals, an age-based subgroup analysis could provide more nuanced insights. Protection against Covid-19 by BNT162b2 Booster across Age Groups
https://www.nejm.org/doi/full/10.1056/NEJMoa2115926 2. Alternative Terminology
To maintain consistency with medical terminology, I recommend using specific clinical terms instead of lay language. This would align better with other variable names mentioned in the manuscript:
Anosmia instead of "loss of sense of smell"
Ageusia instead of "loss of sense of taste"
Anorexia instead of "loss of appetite"
Rhinorrhea and nasal congestion instead of "runny nose" and "stuffy nose"
Dyspnea instead of "shortness of breath"
Author Response
REVIEWER #1
This manuscript explores the impact of recent COVID-19 vaccination on reducing the likelihood of severe COVID-19 among working-age adults. It serves as an extension of outcome analysis from a previous publication. While the methodology is robust and well-executed, the manuscript's content, particularly in the Introduction and Discussion sections, is somewhat underdeveloped. To elevate this work to the level of an original article, I recommend enhancing these sections with additional information and stronger arguments, making it more comprehensive and informative rather than resembling a research letter or note.
Major Concerns. 1. Confounding Variables: Smoking Status Did this study collect smoking status (never smoked, former smoker, current smoker)? Smokers have a higher risk of severe infection compared to those who have never smoked. This could be an important confounding variable to consider in the analysis.
Smoking status was included in the analysis (see second paragraph of data analysis section) but was not identified as a confounder.
2. Confounding Variables: BMI Was BMI data collected, or at least a dichotomous category such as non-obesity vs. obesity? Obese individuals have a significantly higher risk of severe infection compared to those with normal weight. Including this as a confounding factor would strengthen the analysis.
Height and weight were not collected in these surveys of healthy working adults.
3. Immunity Waning and Booster Vaccination The manuscript focuses on how recent COVID-19 vaccinations reduce the severity of infection, but the Introduction and Discussion sections lack depth regarding causes of severe infection, such as "immunity waning." It would be beneficial to include more evidence and discussion on the role of waning immunity and booster vaccination. I recommend expanding these sections using examples from relevant publications: https://www.nejm.org/doi/full/10.1056/NEJMoa2115481 https://www.frontiersin.org/journals/public-health/articles/10.3389/fpubh.2023.1342118/full
Thank you, we have added a paragraph (3rd one) to the introduction introducing information on waning immunity.
Minor Concerns 1. Vaccination Status: Please clarify the vaccination criteria in the Methods section. Did all participants receive at least two doses or complete the primary vaccination series?
No, vaccination status changed over time and cases occurred from day 1 to the end of follow up. As such, the vaccination status of each participant was linked to the date of the positive test. We have tried to make this more clear in the methods section (first paragraph on page 6). Vaccination status ten days prior to the positive test date were used (as also noted in the footer of Table 1).
It’s important that readers are not required to refer back to your previous publication to understand the major inclusion criteria for this study.
The first paragraph of the methods section highlights the eligibility criteria for the parent studies. Additional criteria for this analysis are in the second paragraph. We have added a sentence about vaccination/previous infections, thank you.
Comments 1. Subgroup Analysis by Age Have you considered conducting a subgroup analysis based on age? This could involve categorizing participants by every 10 years or by life stages (e.g., young adults, middle-aged adults, elderly). Alternatively, you could use a simpler dichotomy, such as non-elderly vs. elderly, depending on your outcome analysis. Since older adults tend to experience more severe COVID-19 infections compared to younger individuals, an age-based subgroup analysis could provide more nuanced insights. Protection against Covid-19 by BNT162b2 Booster across Age Groups https://www.nejm.org/doi/full/10.1056/NEJMoa2115926 2.
Thank you. We did not include elderly participants in our cohorts of working-age adults. Also, age was not associated with severity of illness in our regression analyses (i.e., after adjusting for gender, health status, etc.). As such, we did not display sub-group analysis by age. For your information, we re-ran the analyses using age groups, rather than age as a continuous variable, and found no difference in the estimates.
Alternative Terminology To maintain consistency with medical terminology, I recommend using specific clinical terms instead of lay language. This would align better with other variable names mentioned in the manuscript: Anosmia instead of "loss of sense of smell" Ageusia instead of "loss of sense of taste" Anorexia instead of "loss of appetite" Rhinorrhea and nasal congestion instead of "runny nose" and "stuffy nose" Dyspnea instead of "shortness of breath"
We have made these changes, thank you.
Reviewer 2 Report
Comments and Suggestions for Authors
Thank you for inviting me to review the article by Fischer et al. The authors included 3,093 participants and aimed to investigate whether vaccination against SARS-CoV-2 affects the severity of COVID-19 symptoms and whether these symptoms vary by variant. The study concluded that "Recent vaccination against COVID-19 reduces the reported severity of illness in working-age adults who test positive for SARS-CoV-2." However, the symptoms were self-reported by participants, and only healthcare and education workers were included. My concern is about the novelty of these findings, as previous studies on SARS-CoV-2 vaccination and other infections have shown that even if vaccines cannot prevent infection, they can reduce the severity of the disease through partial neutralizing antibodies or T-cell immunity.
Author Response
Thank you for inviting me to review the article by Fischer et al. The authors included 3,093 participants and aimed to investigate whether vaccination against SARS-CoV-2 affects the severity of COVID-19 symptoms and whether these symptoms vary by variant. The study concluded that "Recent vaccination against COVID-19 reduces the reported severity of illness in working-age adults who test positive for SARS-CoV-2." However, the symptoms were self-reported by participants, and only healthcare and education workers were included.
We agree that only healthcare and education workers were included, however the sample sizes were large, included participants from a variety of vocations and age groups, and the duration of the study long. We believe that self-reported symptoms are valid and reliable and the conclusions are valid.
My concern is about the novelty of these findings, as previous studies on SARS-CoV-2 vaccination and other infections have shown that even if vaccines cannot prevent infection, they can reduce the severity of the disease through partial neutralizing antibodies or T-cell immunity.
Our study is one of only very few that use the participant’s subjective assessment of the severity of illness to describe their symptoms. We believe that this study is supportive of other studies reliant on
hospitalized patients and has the added benefit of using data of illnesses caused by many different variants using the same questionnaires and methods.